# Trust the Model When It Is Confident: Masked Model-based Actor-Critic

**Feiyang Pan**[*1,3]    **Jia He**[*2]    **Dandan Tu**[2]    **Qing He**[1,3]

[1]IIP, Institute of Computing Technology, Chinese Academy of Sciences.
[2]Huawei EI Innovation Lab.
[3]University of Chinese Academy of Sciences.
`{pfy824, hejia0149}@gmail.com, tudandan@huawei.com, heqing@ict.ac.cn`

## Abstract

It is a popular belief that model-based Reinforcement Learning (RL) is more sample efficient than model-free RL, but in practice, it is not always true due to overweighed model errors. In complex and noisy settings, model-based RL tends to have trouble using the model if it does not know *when to trust the model*.

In this work, we find that better model usage can make a huge difference. We show theoretically that if the use of model-generated data is restricted to state-action pairs where the model error is small, the performance gap between model and real rollouts can be reduced. It motivates us to use model rollouts only when the model is confident about its predictions. We propose Masked Model-based Actor-Critic (M2AC), a novel policy optimization algorithm that maximizes a model-based lower-bound of the true value function. M2AC implements a masking mechanism based on the model's uncertainty to decide whether its prediction should be used or not. Consequently, the new algorithm tends to give robust policy improvements. Experiments on continuous control benchmarks demonstrate that M2AC has strong performance even when using long model rollouts in very noisy environments, and it significantly outperforms previous state-of-the-art methods.

## 1    Introduction

Deep RL has achieved great successes in complex decision-making problems [17, 21, 9]. Most of the advances are due to deep model-free RL, which can take high-dimensional raw inputs to make decisions without understanding the dynamics of the environment. Although having good asymptotic performance, current model-free RL methods usually require a tremendous number of interactions with the environment to learn a good policy. On the other hand, model-based reinforcement learning (MBRL) enhances sample efficiency by searching the policy under a fitted model that approximates the true dynamics, so it is favored in problems where only a limited number of interactions are available. For example, in continuous control, recently model-based policy optimization (MBPO, [11]) yields comparable results to state-of-the-art model-free methods with much fewer samples, which is appealing for real applications.

However, there is a fundamental concern of MBRL [7] that *learning a good policy requires an accurate model, which in turn requires a large number of interactions with the true environment*. For this issue, theoretical results of MBPO [11] suggest to use the model when the model error is "sufficiently small", which contradicts the intuition that MBRL should be used in low-data scenario. In other words, "*when to trust the model?*" remains an open problem, especially in settings with nonlinear and noisy dynamics and one wants to use a deep neural network as the model.

---

[*]Equal contribution.

Due to the model bias, existing deep MBRL methods tend to perform poorly when 1) the model overfits in low-data regime, 2) the algorithm runs long-horizon model rollouts, and 3) the true dynamic is complex and noisy. As a result, most state-of-the-art MBRL algorithms use very short model rollouts (usually less than four steps [7, 3, 16, 11]), and the applicability is often limited in settings with deterministic dynamics. We show two motivating examples in Figure 1, which demonstrates that MBPO [11], a state-of-the-art MBRL method, indeed suffers from the mentioned limitations. These issues make them impractical to use in real-world problems.

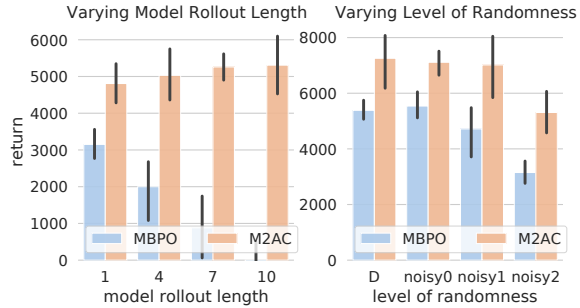

In this work, we introduce Masked Model-based Actor-Critic (M2AC), which alleviates the mentioned issues by reducing large influences of model errors through a masking mechanism. We derive theoretical results that if the model is only used when the model error is small, the gap between the real return and the masked model rollout value can be bounded. This result motivates us to design a practical algorithm M2AC by interpolating masked model-based rollouts with model-free experience replay. We empirically find the new algorithm sample efficient as well as robust across various tasks.

Figure 1: Two motivating examples. Left: returns in HalfCheetah-noisy2 environments at 25k steps. Previous method is sensitive to model rollout length because the model error accumulates as the length increases, while our method (M2AC) can benefit from long rollouts. Right: returns at 25k steps in HalfCheetah and three noisy derivatives with different levels of randomness. The scores of MBPO drop rapidly when the environment becomes noisy, while M2AC is more robust.

Our contributions are outlined as follows.

- We put forward masked model rollouts and derive a general value bounds for Masked Model-based Policy Iteration. We prove that the gap between the model-based and the true Q-values is bounded and can be reduced via better model usage.

- We propose a simple but powerful algorithm named Masked Model-based Actor-Critic (M2AC). It reduces the influences of model error with a masking mechanism that "trusts the model when it is confident" and eliminates unreliable model-generated samples.

- Extensive experiments on continuous control show that M2AC has high sample efficiency that it reaches comparable returns to state-of-the-art model-free methods with much fewer interactions, and is robust even when using long model rollouts in very noisy environments.

## 2 Background

### 2.1 Formulation and notations

We consider an infinite-horizon Markov Decision Process (MDP) $(\mathcal{S}, \mathcal{A}, r, p, p_0, \gamma)$, with $\mathcal{S}$ the state space, $\mathcal{A}$ the action space, $r : \mathcal{S} \times \mathcal{A} \to (-r_{\max}, r_{\max})$ the reward function, $p : \mathcal{S} \times \mathcal{A} \times \mathcal{S} \to \mathbb{R}_{\geq 0}$ the dynamic, $p_0 : \mathcal{S} \to \mathbb{R}_{\geq 0}$ the density of the initial state, and $\gamma \in (0, 1)$ the discount factor. In model-based RL, we need to learn a transition model $\tilde{p}(s'|s, a)$, a reward model $\tilde{r}(s, a)$, and a stochastic policy $\pi : \mathcal{S} \times \mathcal{A} \to \mathbb{R}_{\geq 0}$. For the sake of readability, we make a mild assumption that $r(s, a)$ is *known* so $\tilde{r}(s, a) \equiv r(s, a)$, as it can always be considered as part of the model if unknown. We still use the notation $\tilde{r}$ to make it clear when the reward signal is generated by the learned model.

Given policy $\pi$ and transition $p$, we denote the density of state-action after $t$ steps of transitions from $s$ as $p_t^\pi(s_t, a_t|s)$. Specifically, $p_0^\pi(s, a|s) = \pi(a|s)$. For simplicity, when starting from $p_0(s_0)$, we let $p_t^\pi(s_t, a_t) = \mathbb{E}_{s_0 \sim p_0}[p_t^\pi(s_t, a_t|s_0)]$. Then we have the action-value $Q^\pi(s, a; p) = r(s, a) + \gamma \mathbb{E}_{p(s'|s,a)}[V^\pi(s')]$ and the discounted return $J(\pi; p) = \sum_{t=0}^{\infty} \gamma^t \mathbb{E}_{p_t^\pi}[r(s_t, a_t)]$. We write $Q^\pi(s, a) = Q^\pi(s, a; p)$ and $J(\pi) = J(\pi; p)$ under the true model $p$, and we use $\tilde{Q}^\pi(s, a) = Q^\pi(s, a; \tilde{p})$ and $\tilde{J}(\pi) = J(\pi; \tilde{p})$ under the approximate model $\tilde{p}$.

## 2.2 Previous model-based policy learning methods

The essence of MBRL is to use a learned model $\tilde{p}$ to generate imaginary data so as to optimize the policy $\pi$. Recent studies [16, 11] derive performance bounds in the following form:

$$J(\pi) \geq \tilde{J}(\pi) - D_{p,\pi_b}[(\tilde{p}, \pi), (p, \pi_b)], \tag{1}$$

where $\pi_b$ is an old behavior policy, $D_{p,\pi_b}[(\tilde{p}, \pi), (p, \pi_b)]$ measures some divergence between $(\tilde{p}, \pi)$ and $(p, \pi_b)$. The subscript $p, \pi_b$ means that the divergence is over trajectories induced by $\pi_b$ in the real $p$, i.e., $D_{p,\pi_b}(\tilde{p}, \pi) = \mathbb{E}_{s,a\sim p^{\pi_b}}[f(\tilde{p}, \pi, p, \pi_b; s, a)]$ for some distance measure $f(\cdot)$. SLBO [16] suggests such an error bound, as the expectation is over an old behavior policy in the true dynamics, it can be estimated with samples collected by the behavior policy. Following similar insights, Janner et al. [11] derive a general bound with the following form

$$J(\pi) \geq \tilde{J}(\pi) - c_1 D_{\mathrm{TV}}^{\max}(\pi, \pi_b) - c_2 \delta_{p,\pi_b}^{\max}(\tilde{p}, p). \tag{2}$$

where $D_{\mathrm{TV}}^{\max}(\pi, \pi_b)$ is the divergence between the target policy and the behavior policy, and $\delta_{p,\pi_b}^{\max}(\tilde{p}, p) \triangleq \max_t \mathbb{E}_{s,a\sim p_{t-1}^{\pi_b}}[D_{\mathrm{TV}}[p(s'|s,a), \tilde{p}(s'|s,a)]]$ is the bound of expected model errors induced by a behavior policy $\pi_b$ in the true dynamics $p$ which can be estimated from data.

However, we found two major weaknesses. First, although such results tell us how to *train* the model (i.e., by minimizing $\delta_{p,\pi_b}^{\max}(\tilde{p}, p)$), it does not tell us when to *trust* the model nor how to *use* it. For example, if $\delta_{p,\pi_b}^{\max}(\tilde{p}, p)$ is estimated to be large, which is likely to happen in the beginning of training, should we give up using the model? The second weakness is due to the term of $D_{\mathrm{TV}}^{\max}(\pi, \pi_b)$ which depends on a reference policy. Consider that if we had an almost perfect model (i.e., $\tilde{p} \approx p$), an ideal algorithm should encourage policy search among all possible policies rather than only within the neighborhood of $\pi_b$.

## 2.3 Other related work

Model-based RL has been widely studied for decades for its high sample efficiency and fast convergence [22, 5, 4, 8]. The learned model can be used for planning [4, 24] or for generating imaginary data to accelerate policy learning [22, 5, 8, 7, 3, 16, 18, 11]. Our work falls into the second category.

To improve the policy with model-generated imaginary data in a general non-linear setting, two main challenges are model-learning and model-usage. To obtain a model with less model error, the model ensemble has been found helpful for reducing overfitting [13, 3, 11, 16], which is also used in our method. For better model-usage, except for the mentioned SLBO [16] and MBPO [11], another line of work is model-based value expansion (MVE, [7]) which uses short model rollouts to accelerate model-free value updates. STEVE [3] extends MVE to use stochastic ensemble of model rollouts. Our method also uses stochastic ensemble of imaginary data for policy improvement.

This work is also related to uncertainty estimation in MBRL, which leads to better model-usage if the model "knows what it knows" [15]. To name some, PETS [4] uses the model's uncertainty for stochastic planning. [12] uses imaginary data only when the critic has high uncertainty. STEVE [3] reweights imaginary trajectories according to the variance of target values to stabilize value updates. Our masking method can also be understood as reweighting with binary weights, but we only measure uncertainty in the dynamics thus does not depend on the actor nor the critic.

## 3 Masked Model-based Policy Iteration

### 3.1 Model-based return bounds

To guide a better model usage, we seek to derive a general performance guarantee for model-based policy learning in the following "model-based" form without relying on any old reference policy

$$J(\pi) \geq \tilde{J}(\pi) - D_{\tilde{p},\pi}(\tilde{p}, p). \tag{3}$$

To get it, let $\delta_{\tilde{p},\pi}^{(t)}(\tilde{p}, p) \triangleq \mathbb{E}_{s,a\sim \tilde{p}_{t-1}^{\pi}}[D_{\mathrm{TV}}[p(s'|s,a), \tilde{p}(s'|s,a)]]$ be the expected model error at step $t$, $t \geq 1$, where the expectation is over the state-action pairs encountered by running the *new* policy $\pi$ in the *fitted* model $\tilde{p}$. Then if the model rollouts starts at $p_0(s_0)$, we have

**Theorem 1** (Model-based performance bound for model-based policy iteration)**.**

$$J(\pi) \geq \tilde{J}(\pi) - \frac{2r_{\max}}{1-\gamma} \sum_{t=1}^{+\infty} \gamma^t \delta_{\tilde{p},\pi}^{(t)}(\tilde{p}, p) \tag{4}$$

The proof can be found in Appendix B. Comparing to Eq. (2), a clear drawback of Eq. (4) is that the cumulative model error cannot be estimated by observed data and it can be large for unconstrained long model rollouts. To reduce the discrepancy, we wish to construct a new bound by changing the way we use the model. Intuitively, if we could limit the use of the model to states where the model error was small, the return discrepancy could be smaller. To this end, we define *masked model rollouts*, which is not a real rollout method but is useful for theoretical analysis.

## 3.2 Masked model rollouts

To formulate the masked model rollout, we introduce a few more notations. Imagine a hybrid model that could switch between using the fitted model and the true dynamics, and a binary *mask* $M : \mathcal{S} \times \mathcal{A} \to \{0, 1\}$ over state-action pairs. Consider that the model rollout starts at state $s_0$. Once it encounters a state-action pair that has $M(s_H, a_H) = 0$, it switches to use the true dynamics. Here we use the uppercase $H$ as a random variable of the stopping time. So the transition becomes $p_{\text{mask}}(s_{t+1}|s_t, a_t) = \mathbb{I}\{t < H\}\tilde{p}(s_{t+1}|s_t, a_t) + \mathbb{I}\{t \geq H\}p(s_{t+1}|s_t, a_t)$. Such a masked rollout trajectory has the following form (given model $\tilde{p}$, mask $M$, policy $\pi$, and state $s_0$)

$$\tau \mid \tilde{p}, \pi, M, s_0 : s_0, \underbrace{a_0 \xrightarrow{\tilde{p}} \cdots \xrightarrow{\tilde{p}} s_H}_{\text{rollout with } (\tilde{p}, \pi)}, \underbrace{a_H \xrightarrow{p} s_{H+1}, a_{H+1} \xrightarrow{p} \cdots}_{\text{rollout with } (p, \pi)}$$

Then we define $\tilde{Q}_{\text{mask}}(s, a) = \mathbb{E}_{\tau, H}[\sum_{t=0}^{H-1} \gamma^t \tilde{r}(s, a) + \sum_{t=H}^{\infty} \gamma^t r(s, a) \mid \tilde{p}, \pi, M, s_0 = s, a_0 = a]$.

## 3.3 Return bounds for masked model policy iteration

Given a model $\tilde{p}$ and a binary mask $M$, let $\epsilon = \max_{s,a} M(s, a) D_{\text{TV}}[p(s'|s, a), \tilde{p}(s'|s, a)]$ be the maximum model error over masked state-action pairs. Then we have the following bound

**Theorem 2.** *Given $\tilde{p}$ and $M$, the Q-value of any policy $\pi$ in the real environment is bounded by*

$$Q^\pi(s, a) \geq \tilde{Q}_{mask}^\pi(s, a) - \alpha\epsilon \sum_{t=0}^{+\infty} \gamma^t w(t; s, a), \tag{5}$$

*where $\alpha = \frac{2r_{\max}}{1-\gamma}$, and $w(t; s, a) = \Pr\{t < H \mid \tilde{p}, \pi, M, s_0 = s, a_0 = a\}$ is the expected proportion of model-based transition at step $t$.*

The proof is in Appendix B. We can extend this result that if the model rollout length is limited to $H_{\max}$ steps, then $|Q^\pi(s, a) - \tilde{Q}_{\text{mask}}^\pi(s, a)| \leq \alpha\epsilon \sum_{t=0}^{H_{\max}-1} \gamma^t w(t; s, a)$, which grows sub-linearly with $H_{\max}$. Therefore, for each $(s, a)$ that $M(s, a) = 1$, to maximize $Q^\pi(s, a)$, we can instead maximize its lower-bound, which can be rewritten as

$$Q^\pi(s, a) \geq \mathbb{E}_{\tau, H}\left[ \sum_{t=0}^{H-1} \gamma^t(\tilde{r}(s_t, a_t) - \alpha\epsilon) + \gamma^H Q^\pi(s_H, a_H) \,\middle|\, \tilde{p}, \pi, M, s, a \right] \tag{6}$$

So given the model $(\tilde{p}, \tilde{r})$, the bound depends on $M$ and $\pi$, which corresponds to *model usage* and *policy optimization*, respectively. To design a practical algorithm, we conclude the following insights.

First, the return for each trajectory is the sum of 1) a model-based $H$-step return subtracted by a penalty of model-bias, and 2) a model-free long-term return. For the first part, as it is model-based, it is always tractable. For the second part, as it is a model-free $Q$-value, we could use model-free RL techniques to takle it. Therefore, we can use a combination of model-based and model-free RL to approximate the masked model rollout.

Second, the results tell us that the gap between model returns and actual returns depend on the model bias in model rollouts instead of that in real rollouts. Standard model-based methods can be

understood as $M(s, a) \equiv 1$ for all $(s, a)$ during model rollouts though the model error can be large (or even undefined, because the imaginary state may not be a valid state), so they suffer from large performance gap especially when using long model rollouts. It suggests to restrict model usage to reduce the gap, which leads to our actual algorithm.

Third, as written in Eq. (5), the gap is in the form of $\epsilon \cdot w$. As $\epsilon$ is the maximum error during model rollouts, it is difficult to quantitatively constrain $\epsilon$, if possible. However, we can always control $w$, the proportion of model-based transitions to use. It motivates our algorithm design with a rank-based heuristic, detailed in the next section.

# 4  Masked Model-based Actor-Critic

Following the theoretical results, we propose Masked Model-based Actor-Critic (M2AC).

---
**Algorithm 1** Actual algorithm of M2AC
---
1: **Inputs:** $K, B, H_{\max}, w, \alpha$
2: Random initialize policy $\pi_\phi$, Q-function $Q_\psi$, and $K$ predictive models $\{p_{\theta_1}, \ldots, p_{\theta_K}\}$.
3: **for** each epoch **do**
4:     Collect data with $\pi$ in real environment: $\mathcal{D}_{\text{env}} = \mathcal{D}_{\text{env}} \cup \{(s_i, a_i, r_i, s_i')\}_i$
5:     Train models $\{p_{\theta_1}, \ldots, p_{\theta_K}\}$ on $\mathcal{D}_{\text{env}}$ with early stopping
6:     Randomly sample $B$ states from $\mathcal{D}_{\text{env}}$ with replacement: $\boldsymbol{S} \leftarrow \{s_i\}_i^B$
7:     $\mathcal{D}_{\text{model}} \leftarrow \text{MaskedModelRollouts}(\pi_\phi, \{p_{\theta_1}, \ldots, p_{\theta_K}\}, \boldsymbol{S}; H_{\max}, w, \alpha)$
8:     Update $\pi_\phi$ and $Q_\psi$ with off-policy actor-critic algorithm on experience buffer $\mathcal{D}_{\text{env}} \cup \mathcal{D}_{\text{model}}$
---

## 4.1  Actor-Critic with function approximation

To replace the Q-values in (6) with practical function approximation, we first define

$$\tilde{Q}_{\text{M2AC}}^\pi(s, a) = \begin{cases} r(s, a) + \gamma \mathbb{E}[\tilde{Q}_{\text{M2AC}}^\pi(s', a') \mid s', a' \sim p, \pi], & M(s, a) = 0, \\ \tilde{r}(s, a) - \alpha\epsilon + \gamma \mathbb{E}[\tilde{Q}_{\text{M2AC}}^\pi(s', a') \mid s', a' \sim \tilde{p}, \pi], & M(s, a) = 1. \end{cases} \tag{7}$$

From Theorem 2, it is easy to see that $\tilde{Q}_{\text{M2AC}}^\pi(s, a)$ is a lower-bound of the true Q-function.

**Corollary 1.** $\tilde{Q}_{M2AC}^\pi(s, a) \leq Q^\pi(s, a)$ *for all* $(s, a)$.

The recursive form of Eq. (7) indicates that, when $M(s, a) = 0$, we should use standard model-free Q-update with samples of real transitions $(s, a, r, s')$; when $M(s, a) = 1$, we can perform a model-based update with model-generated data $(s, a, \tilde{r} - \alpha\epsilon, \tilde{s}')$. Such a hybrid Temporal Difference learning can be performed directly via experience replay [2]. As noted in Algorithm 1, we use an experience buffer $\mathcal{D}_{\text{env}} \cup \mathcal{D}_{\text{model}}$, where $\mathcal{D}_{\text{model}}$ is generated with model rollouts (detailed in Algorithm 2). In particular, as the maximum model error $\epsilon$ is intractable to obtain, we follow a similar technique of TRPO [19] to replace it by a sample-based estimation of model error $u(s, a)$, detailed in Section 4.3. Then we can use off-the-shelf actor-critic algorithms to update the Q-functions as well as the policy. In our implementation, we use Soft-Actor-Critic (SAC) [10] as the base algorithm.

## 4.2  The masking mechanism

Designing a reasonable masking mechanism plays a key role in our method. On the one hand, given a model $\tilde{p}$, we should restrict the mask to only exploit a small "safe zone" so that $\epsilon$ can be small. On the other hand, the mask should not be too restricted, otherwise the agent can hardly learn anything (e.g., $M(s, a) \equiv 0$ turns off the model-based part).

We propose a simple rank-based filtering heuristic as the masking mechanism. Consider that the agent runs a minibatch of model rollouts in parallel and has generated a minibatch of transitions $\{(s_i, a_i, r_i, s_i')\}_i^B$, where each transition is assigned with an uncertainty score $u_i \approx D_{\text{TV}}[\tilde{p}(\cdot|s_i, a_i), p(\cdot|s_i, a_i)]$ as the estimated model error at input $(s_i, a_i)$. We rank these rollout samples by their uncertainty score, and only keep the first $wB$ samples with small uncertainty. Here $w \in (0, 1]$ is a hyper-parameter of the masking rate. The overall model-based data generation process is outlined in Algorithm 2.

Such a masking mechanism has a few advantages. First, by setting this proportion we can control the performance gap in Eq. (5) (although not controlled explicitly, the term $w(t; s, a)$ in average would be close to $w$). Second, as it is rank-based, we only need to find a predictor $u(s, a)$ that is positively correlated to $D_{\text{TV}}[\tilde{p}(s', r|s, a), p(s', r|s, a)]$. Last but not least, it is simple and easy to implement.

Specifically, we have two rollout modes, the hard-stop mode and the non-stop mode, illustrated in Figure 2. Both modes rollout for at most $H_{\max}$ steps from the initial states. Hard-stop mode stops model rollouts once it encounters an $(s, a)$ that $M(s, a) = 0$; Non-stop mode always runs $H_{\max}$ steps and only keeps the samples that has $M(s, a) = 1$.

---

**Algorithm 2** Masked Model Rollouts for data generation

1: **Inputs:** policy $\pi$, $K$ models $\{p_{\theta_1}, \ldots, p_{\theta_K}\}$, initial states $\boldsymbol{S}$, max rollout length $H_{\max}$, masking rate $w$, penalty coef. $\alpha$
2: Initialize $B \leftarrow |\boldsymbol{S}|$, $\mathcal{D}_{\text{model}} \leftarrow \emptyset$.
3: **for** $h = 0, \ldots, H_{\max} - 1$ **do**
4:     **for** $i = 1, \ldots, B$ **do**
5:         Choose action $a_i \leftarrow \pi(s_i)$
6:         Compute $p_{\theta_k}(r, s'|s_i, a_i)$ for $k = 1, \ldots, K$
7:         Randomly choose $k$ from $1, \ldots, K$
8:         Draw $\tilde{r}_i, s'_i \sim p_{\theta_k}(r, s'|s_i, a_i)$
9:         Compute One-vs-Rest uncertainty estimation $u_i$ by (9)
10:    Rank the samples by $u_i$. Get $\lfloor wB \rfloor$ samples' indexes $\{i_j\}_j^{\lfloor wB \rfloor}$ with smaller uncertainty scores
11:    $\mathcal{D}_{\text{model}} \leftarrow \mathcal{D}_{\text{model}} \cup \{(s_{i_j}, a_{i_j}, \tilde{r}_{i_j} - \alpha u_{i_j}, s'_{i_j})\}_j^{\lfloor wB \rfloor}$
12:    (Non-stop mode) $\boldsymbol{S} \leftarrow \{s'_i\}_i^B$
13:    (Hard-stop mode, *optional*) $\boldsymbol{S} \leftarrow \{s_{i_j}\}_j^{\lfloor wB \rfloor}$; $B \leftarrow \lfloor wB \rfloor$
14: **Return:** $\mathcal{D}_{\text{model}}$

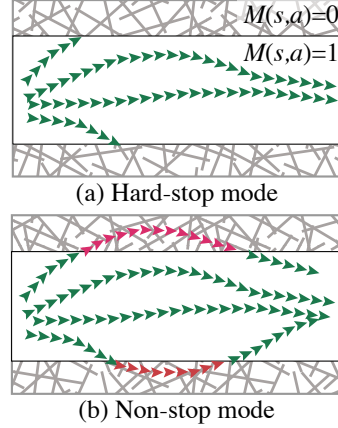

(a) Hard-stop mode

(b) Non-stop mode

Figure 2: Rollout modes in M2AC.

---

## 4.3 Probabilistic model ensemble and model error prediction

Now the only obstacle is how to train a model that can provide the uncertainty estimation. There are three requirements. First, to deal with stochastic dynamics, it should be able to learn the *aleatoric* uncertainty in the dynamics. Second, to prevent overfitting in low-data scenario, it should keep track of the *epistemic* uncertainty (aka the model uncertainty). Finally, the model should be capable of predicting an uncertainty score $u(s, a)$ that is positively correlated to $D_{\text{TV}}[\tilde{p}(s', r|s, a), p(s', r|s, a)]$.

In our work, we use an ensemble of probabilistic neural networks as the model. Similar to [13, 3, 11], we train a bag of $K$ models $\{p_{\theta_1}, \cdots, p_{\theta_K}\}$, $K \geq 2$. Each model $p_{\theta_k}(r, s'|s, a)$ is a probabilistic neural network that outputs a distribution over $(r, s')$. All the models are trained in parallel with different data shuffling and random initialization. For inference, the agent randomly chooses one model $p_{\theta_k}$ to make a prediction. In this way, both aleatoric and model uncertainties can be learned.

Then we propose One-vs-Rest (OvR) uncertainty estimation, a simple method to estimate model errors with model ensembles. Given $(s, a)$ and a chosen model index $k$, we define the estimator as:

$$u_k(s, a) = D_{\text{KL}}[p_{\theta_k}(\cdot|s, a) \| p_{-k}(\cdot|s, a)], \tag{8}$$

where $p_{-k}(s', r|s, a)$ is the ensembled prediction of the other $K - 1$ models except model $k$. It measures the disagreement between one model versus the rest of the models. Therefore, a small $u_k(s, a)$ indicates that the model is confident about its prediction.

As a concrete example, for tasks with continuous states and actions, we set the model output as diagonal Gaussian, i.e., $p_{\theta_i}(s', r|s, a) = \mathcal{N}(\mu_{\theta_i}(s, a), \sigma^2_{\theta_i}(s, a))$. We use negative log-likelihood (NLL) as the loss function for training as well as for early-stopping on a hold-out validation set. For OvR uncertainty estimation, we also use diagonal Gaussian as $p_{-k}(s', r|s, a)$ by merging multiple Gaussian distributions, i.e., $p_{\theta_{-k}}(s', r|s, a) = \mathcal{N}(\mu_{-k}(s, a), \sigma^2_{-k}(s, a))$ with $\mu_{-k}(s, a) = \frac{1}{K-1} \sum_{i \neq k}^K \mu_{\theta_i}(s, a)$ and $\sigma^2_{-k}(s, a) = \frac{1}{K-1} \sum_{i \neq k}^K (\sigma^2_{\theta_i}(s, a) + \mu^2_{\theta_i}(s, a)) - \mu^2_{-k}(s, a)$. Then it is simple to compute the KL-divergence between two Gaussians as the uncertainty score

$$u_k(s, a) = D_{\text{KL}}[\mathcal{N}(\mu_{\theta_i}(s, a), \sigma^2_{\theta_i}(s, a)) \| \mathcal{N}(\mu_{-k}(s, a), \sigma^2_{-k}(s, a))]. \tag{9}$$

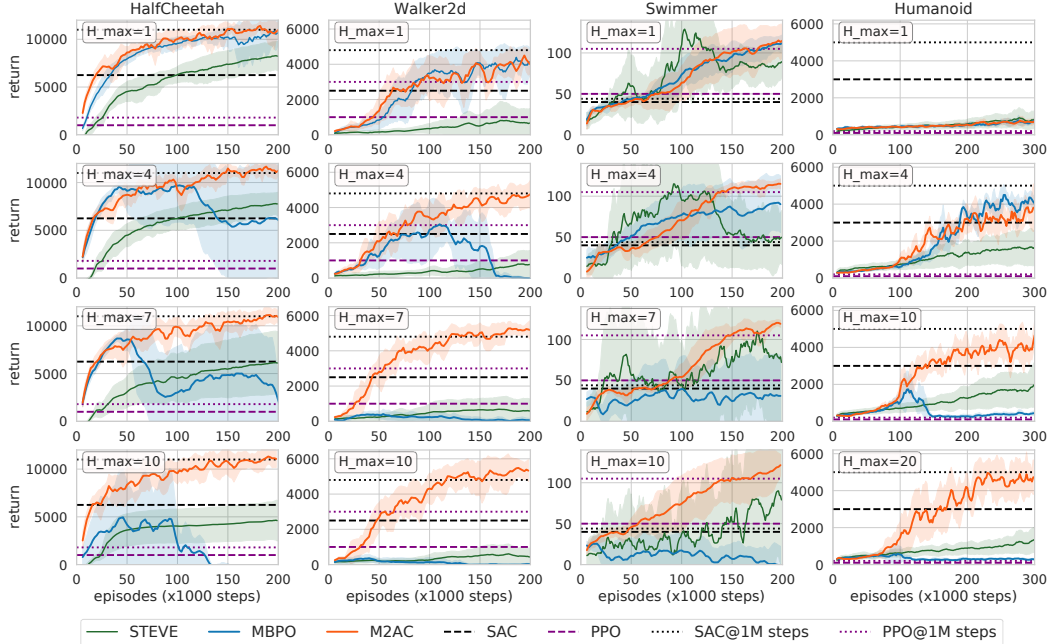

Figure 3: Training curves on MuJoCo-v2 benchmarks. Solid curves are average returns over 7 runs, and shaded areas indicate one standard deviation. The dashed lines are the returns of SAC and PPO at the maximum training steps (200k steps for the first three tasks, and 300k steps for Humanoid-v2), and the dotted lines are the returns at 1 million steps (retrived from the original papers [9, 20] and opensourced benchmarks [6, 1]).

## 5 Experiments

Our experiments are designed to investigate two questions: (1) How does M2AC compare to prior model-free and model-based methods in sample efficiency? (2) Is M2AC stable and robust across various settings? How does its robustness compare to existing model-based methods?

### 5.1 Baselines and implementation

Our baselines include model-free methods PPO [20] and SAC [9], and two state-of-the-art model-based methods STEVE [3] and MBPO [11]. In the implementation of MBRL methods, the most important hyper-parameter is the maximum model rollout length $H_{\max}$. In MBPO [11] the authors test and tune this parameter for every task, but we argue that in real applications we can not know the best choice in advance nor can we tune it. So for a fair and comprehensive comparison, we test $H_{\max} = 1, 4, 7, 10$ for all the algorithms and tasks.

To ensure a fair comparison, we run M2AC and MBPO with the same network architectures and training configurations based on the opensourced implementation of MBPO (detailed in the appendix). For the comparison with STEVE, there are many different compounding factors (e.g., M2AC and MBPO use SAC as the policy optimizer while STEVE uses DDPG) so we just test it as is.

The main experiments involve the following implementation details for M2AC: the masking uses the non-stop mode (Algorithm 2, line 12). The masking rate is set as $w = 0.5$ when $H_{\max} = 1$, and is a decaying linear function $w_h = \frac{H_{\max} - h}{2(H_{\max} + 1)}$ when $H_{\max} > 1$. The model error penalty is set as $\alpha = 10^{-3}$. Discussions and ablation studies about these parameters are in Section 5.4.

### 5.2 Experiments on continuous control benchmarks with deterministic dynamics

Figure 3 demonstrates the results in four MuJoCo-v2 [23] environments. We observe that M2AC can consistently achieve a high performance with a small number of interactions. Specifically, M2AC performs comparably to the best model-free method with about $5\times$ fewer samples.

Comparing to MBPO, when using $H_{\max} = 1$ (recommended by MBPO [11]), M2AC achieves a comparable score to MBPO, which is easy to understand that the model is almost certain about its prediction in such a short rollout. When $H_{\max} > 1$, we observe that the performance of MBPO drops rapidly due to accumulated model errors. On the contrary, M2AC can even benefit from longer model rollout, which is appealing in real applications.

Comparing to STEVE, we see that M2AC can achieve a higher return (except in Swimmer, because STEVE bases on DDPG while M2AC bases on SAC and it is known that DDPG performs much better in Swimmer than SAC), and is significantly more stable. Notably, although STEVE seems not as competitive as MBPO in their best parameters, it outperforms MBPO when using long rollout lengths, most likely because of its reweighting mechanism. Therefore, we conclude that using uncertainty-based technique indeed enhances generalization in long model rollouts, and our M2AC is clearly more reliable than all the competitors in the tested benchmarks.

### 5.3 Experiments in noisy environments

In order to understand the behavior and robustness of MBRL methods in a more realistic setting, in this section we conduct experiments of noisy environments with a very few interactions, which is challenging for modern deep RL. The noisy environments are set-up by adding noises to the agent's action at every step, i.e., each step the environment takes $a' = a + \mathrm{WN}(\sigma^2)$ as the action, where $a$ is the agent's raw action and $\mathrm{WN}(\sigma^2)$ is an unobservable Gaussian white noise with $\sigma$ its standard deviation. We implement noisy derivatives based on HalfCheetah and Walker2d, each with three levels: $\sigma = 0.05$ for "-Noisy0", $\sigma = 0.1$ for "-Noisy1", and $\sigma = 0.2$ for "-Noisy2". Since STEVE is not so competitive in the deterministic tasks, we do not include it for comparison.

The results are shown in Figure 4. Despite the noisy dynamics are naturally harder to learn, we see that M2AC performs robustly in all the environments and significantly outperforms MBPO. In HalfCheetah-based tasks, M2AC is robust to the rollout length even in the most difficult Noisy2 environment. In Walker2d-based tasks, we observe that the return of M2AC also drops when the rollout length is longer. We attribute the reason to the fact that Walker2d is known to be more difficult than HalfCheetah, so the performance of a MBRL algorithm in Walker2d is more sensitive to model errors. But overall, comparing to MBPO which can merely learn anything from noisy Walker2d environments, M2AC is significantly more robust.

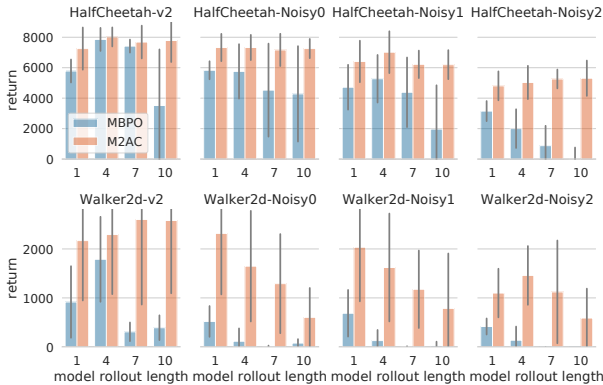 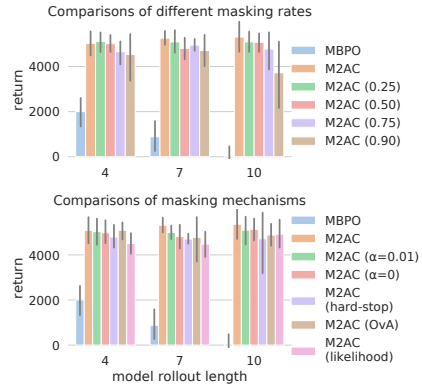

Figure 4: Results in noisy environments with very few interactions (25k steps for HalfCheetah and 50k steps for Walker2d). The left-most column is the deterministic benchmarks, the other three columns are the noisy derivatives. The bars are average returns over 10 runs and error lines indicate one standard deviation.

Figure 5: Ablation studies (in HalfCheetah-Noisy2). Up: comparisons between masking rates. Down: comparisons of masking mechanisms. Error bars are 90% bootstrap confidence intervals.

### 5.4 Ablation studies

Our first ablation study is about the most important hyper-parameter, the **masking rate** $w$. The result is shown in the upper figure of Figure 5. It show that, surprisingly, our method is not so

sensitive to the choice of $w$. Only setting $w$ too close to one ($w = 0.9$) in long model rollouts degrades the performance, indicating that smaller $w$ is better when the rollout step increases. Also, we see that $w = 0.25$ works fairly well. Therefore, we recommend the linear decaying masking rate $w_h = \frac{H_{\max} - h}{2(H_{\max}+1)}$ whose average $\frac{1}{H_{\max}} \sum_{h=0}^{H_{\max}-1} w_h$ is 0.25. We observe that it performs robustly across almost all settings.

Then we conducted three ablation studies about the masking mechanism, which are demonstrated together in the lower figure of Figure 5.

On the **penalty coefficient** $\alpha$, we observe that M2AC with $\alpha = 0.001$ performs the best among $[0.01, 0.001, 0]$. To explain, too small an $\alpha$ encourages model exploitation, i.e., the policy can overfit to the model. Too large an $\alpha$ encourages exploration in unfamiliar areas of the real environment but discourages the use of the model. Therefore, a moderate choice such as $\alpha = 0.001$ could be better.

On the **multi-step masked rollout mode**, we find the non-stop mode better than the hard-stop mode (while the later runs faster), mainly because the non-stop mode leads to a substantially richer distribution of imaginary transitions, which enhances the generalization power of the actor-critic part.

Finally, for **uncertainty estimation**, we compare two other methods: 1) One-vs-All (OvA) disagreement [14], which simply replaces the leave-one-out ensemble in our OvR estimation with a complete ensemble of $K$ models. 2) Negative likelihood, which estimates the likelihood of the sampled transition $(s, a, \tilde{r}, \tilde{s}')$ in the predicted ensemble distribution as a measure of confidence. The results show that our OvR uncertainty estimation outperforms the others. We attribute it to that OvR is better at measuring the disagreement between the models, which is crucial in low-data regimes.

# 6    Conclusion

In this paper, we derive a general performance bound for model-based RL and theoretically show that the divergence between the return in the model rollouts and that in the real environment can be reduced with restricted model usage. Then we propose Masked Model-based Actor-Critic which incorporates model-based data generation with model-free experience replay by using the imaginary data with low uncertainty. By extensive experiments on continuous control benchmarks, we show that M2AC significantly outperforms state-of-the-art methods with enhanced stability and robustness even in challenging noisy tasks with long model rollouts. Further ablation studies show that the proposed method is not sensitive to its hyper-parameters, thus it is reliable to use in realistic scenarios.

## Broader impact

This paper aims to make reinforcement learning more reliable in practical use. We point out that previous methods that work well under ideal conditions can have a poor performance in a more realistic setting (e.g., in a noisy environment) and are not robust to some hyper-parameters (e.g., the model rollout length). We suggest that these factors should be taken into account for future work. We improve the robustness and generalization ability by restricting model use with a notion of uncertainty. Such an insight can be universal in all areas for building reliable Artificial Intelligence systems.

## Acknowledgements

The research is supported by National Key Research and Development Program of China under Grant No. 2017YFB1002104, National Natural Science Foundation of China under Grant No. U1811461.

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
