[Supplementary Material]

# A Appendix: Lemmas

**Lemma 1** (Total-variance divergence of joint distributions). *Given two joint distributions $p(x,y) = p(y|x)p(x)$ and $\tilde{p}(x,y) = \tilde{p}(y|x)\tilde{p}(x)$, the total-variance divergence can be bounded by*

$$D_{TV}[p(x,y), \tilde{p}(x,y)] \leq D_{TV}[p(x), \tilde{p}(x)] + \mathbb{E}_{x \sim \tilde{p}}[D_{TV}[p(y|x), \tilde{p}(y|x)]] \tag{10}$$

*Proof.*

$$
\begin{aligned}
D_{\mathrm{TV}}[p(x,y), \tilde{p}(x,y)] &= \frac{1}{2} \sum_{x,y} |p(x,y) - \tilde{p}(x,y)| \\
&= \frac{1}{2} \sum_{x,y} |p(y|x)p(x) - \tilde{p}(y|x)\tilde{p}(x)| \\
&= \frac{1}{2} \sum_{x,y} |p(y|x)p(x) - p(y|x)\tilde{p}(x) + p(y|x)\tilde{p}(x) - \tilde{p}(y|x)\tilde{p}(x)| \\
&\leq \frac{1}{2} \sum_{x,y} p(y|x)|p(x) - \tilde{p}(x)| + \frac{1}{2} \sum_{x,y} \tilde{p}(x)|p(y|x) - \tilde{p}(y|x)| \\
&= \frac{1}{2} \sum_{x} |p(x) - \tilde{p}(x)| + \frac{1}{2} \sum_{x} \tilde{p}(x) \sum_{y} |p(y|x) - \tilde{p}(y|x)| \\
&= D_{\mathrm{TV}}[p(x), \tilde{p}(x)] + \mathbb{E}_{x \sim \tilde{p}}[D_{\mathrm{TV}}[p(y|x), \tilde{p}(y|x)]]
\end{aligned}
$$

$\square$

**Lemma 2** (Total-variance divergence of two Markov chains). *Given two Markov chain $p_{t+1}(x') = \sum_x p(x'|x)p_t(x)$ and $\tilde{p}_{t+1}(x') = \sum_x \tilde{p}(x'|x)\tilde{p}_t(x)$, we have*

$$D_{TV}[p_{t+1}(x'), \tilde{p}_{t+1}(x')] \leq D_{TV}[p_t(x), \tilde{p}_t(x)] + \mathbb{E}_{x \sim \tilde{p}_t}[D_{TV}[p(x'|x), \tilde{p}(x'|x)]]. \tag{11}$$

Given policy $\pi$ and transition $p$, we denote the density of state-action after $t$ steps of transitions from $s$ as $p_t^\pi(s_t, a_t|s)$. Specifically, $p_0^\pi(s, a|s) = \pi(a|s)$. For simplicity, when starting from $p_0(s_0)$, we let $p_t^\pi(s_t, a_t) = \mathbb{E}_{s_0 \sim p_0}[p_t^\pi(s_t, a_t|s_0)]$.

**Lemma 3** (Return gap). *Consider running two policies of $\pi$ and $\tilde{\pi}$ in two dynamics $p$ and $\tilde{p}$, respectively. Let $D_{TV}^{\max}(\pi, \tilde{\pi}) \triangleq \max_s D_{TV}[\pi_b(\cdot|s), \pi(\cdot|s)]$, $\delta^{(0)}(\tilde{p}, p) \triangleq D_{TV}[p_0(s), \tilde{p}_0(s)]$, $\delta_{\tilde{p},\tilde{\pi}}^{(t)}(\tilde{p}, p) \triangleq \mathbb{E}_{s,a \sim \tilde{p}_{t-1}^{\tilde{\pi}}}[D_{TV}[p(s'|s,a), \tilde{p}(s'|s,a)]]$ for $t \geq 1$. Then the discounted returns up to step $T$ are bounded as*

$$|J_T(\pi) - \tilde{J}_T(\tilde{\pi})| \leq 2r_{\max}\left(\frac{1}{(1-\gamma)^2} D_{TV}^{\max}(\pi, \tilde{\pi}) + \frac{1}{1-\gamma} \sum_{t=0}^{T} \gamma^t \delta_{\tilde{p},\tilde{\pi}}^{(t)}(\tilde{p}, p)\right). \tag{12}$$

*Proof.* The expected rewards at step $t$ is bounded by

$$|\mathbb{E}_{p_t^\pi}[r(s,a)] - \mathbb{E}_{\tilde{p}_t^{\tilde{\pi}}}[r(s,a)]| = \Big|\sum_{s,a} r(s,a)(p_t^\pi(s,a) - \tilde{p}_t^{\tilde{\pi}}(s,a))\Big| \tag{13}$$

$$\leq \sum_{s,a} r_{\max}|p_t^\pi(s,a) - \tilde{p}_t^{\tilde{\pi}}(s,a)| \tag{14}$$

$$= 2r_{\max} D_{\mathrm{TV}}[p_t^\pi(s,a), \tilde{p}_t^{\tilde{\pi}}(s,a)] \tag{15}$$

So

$$|J_T(\pi) - \tilde{J}_T(\tilde{\pi})| \leq \sum_{t=0}^{T} \gamma^t |\mathbb{E}_{p_t^\pi}[r(s,a)] - \mathbb{E}_{\tilde{p}_t^{\tilde{\pi}}}[r(s,a)]| \tag{16}$$

$$\leq 2r_{\max} \sum_{t=0}^{T} \gamma^t D_{\mathrm{TV}}[p_t^\pi(s,a), \tilde{p}_t^{\tilde{\pi}}(s,a)] \tag{17}$$

By Lemma 1 and Lemma 2, we have

$$D_{\text{TV}}[p_t^\pi(s,a), \tilde{p}_t^{\tilde{\pi}}(s,a)] \leq D_{\text{TV}}[p_{t-1}^\pi(s,a), \tilde{p}_{t-1}^{\tilde{\pi}}(s,a)] + \delta_{\tilde{p},\tilde{\pi}}^{(t)}(\tilde{p},p) + D_{\text{TV}}^{\max}(\pi,\tilde{\pi}). \tag{18}$$

$$|J_T(\pi) - \tilde{J}_T(\tilde{\pi})| \tag{19}$$

$$\leq 2r_{\max} \sum_{t=0}^{T} \gamma^t \left( (t+1) D_{\text{TV}}^{\max}(\pi,\tilde{\pi}) + \sum_{\tau=0}^{t} \delta_{\tilde{p},\tilde{\pi}}^{(\tau)}(\tilde{p},p) \right) \tag{20}$$

$$= 2r_{\max} \sum_{t=0}^{T} (t+1)\gamma^t D_{\text{TV}}^{\max}(\pi,\tilde{\pi}) + 2r_{\max} \sum_{t=0}^{T} \left( \sum_{\tau=t}^{T} \gamma^\tau \right) \delta_{\tilde{p},\tilde{\pi}}^{(t)}(\tilde{p},p) \tag{21}$$

$$= 2r_{\max} \left[ \left( \frac{1-\gamma^{T+1}}{(1-\gamma)^2} - \frac{(T+1)\gamma^{T+1}}{1-\gamma} \right) D_{\text{TV}}^{\max}(\pi,\tilde{\pi}) + \sum_{t=0}^{T} \frac{\gamma^t(1-\gamma^{T-t+1})}{1-\gamma} \delta_{\tilde{p},\tilde{\pi}}^{(t)}(\tilde{p},p) \right] \tag{22}$$

$$\leq \frac{2r_{\max}}{(1-\gamma)^2} D_{\text{TV}}^{\max}(\pi,\tilde{\pi}) + \frac{2r_{\max}}{1-\gamma} \sum_{t=0}^{T} \gamma^t \delta_{\tilde{p},\tilde{\pi}}^{(t)}(\tilde{p},p) \tag{23}$$

$$\square$$

## B   Appendix: Proofs of Theorems

**Theorem 3** (General return bound for model-based policy iteration). $J(\pi)$ *is lower-bounded by*

$$J(\pi) \geq \tilde{J}(\pi) - \frac{2r_{\max}}{1-\gamma} \sum_{t=0}^{+\infty} \gamma^t \delta_{\tilde{p},\pi}^{(t)}(\tilde{p},p).$$

*Proof.* According to Lemma 3, we have

$$J_T(\pi) \geq \tilde{J}_T(\tilde{\pi}) - \frac{2r_{\max}}{(1-\gamma)^2} D_{\text{TV}}^{\max}(\pi,\tilde{\pi}) - \frac{2r_{\max}}{1-\gamma} \sum_{t=0}^{+\infty} \gamma^t \delta_{\tilde{p},\tilde{\pi}}^{(t)}(\tilde{p},p). \tag{24}$$

Since $D_{\text{TV}}^{\max}(\pi,\pi) = 0$, we have the desired result. $\square$

**Theorem 4.** *Given a policy $\pi$ and a state-action pair $(s,a)$, the discrepancy between the long-term return in the real environment and the return in the masked model rollout of $(\tilde{p}, M)$ is bounded by*

$$|Q^\pi(s,a) - \tilde{Q}_{mask}^\pi(s,a)| \leq \epsilon \sum_{t=0}^{+\infty} \gamma^t w(t;s,a),$$

*Proof.* By the definition of masked model rollout, we can write the transition as

$$p_{\text{mask}}(s_{t+1}|s_t,a_t) = \mathbb{I}\{t < H\}\tilde{p}(s_{t+1}|s_t,a_t) + \mathbb{I}\{t \geq H\}p(s_{t+1}|s_t,a_t).$$

So we can bound the single-step model error by

$$\mathbb{E}_\tau[D_{\text{TV}}[p(s_{t+1}|s_t,a_t), p_{\text{mask}}(s_{t+1}|s_t,a_t)] \mid \tilde{p}, \pi, M, s_0 = s, a_0 = a] \tag{25}$$

$$= \mathbb{E}_\tau[\mathbb{I}\{t < H\}D_{\text{TV}}[p(s_{t+1}|s_t,a_t), \tilde{p}(s_{t+1}|s_t,a_t)] \mid \tilde{p}, \pi, M, s_0 = s, a_0 = a] \tag{26}$$

$$\leq \epsilon w(t;s,a) \tag{27}$$

So by Lemma 3, we have the desired result. $\square$

## C   Appendix: Hyper-parameters

- Environment steps per epoch: 1000.
- Policy updates per epoch: 10000.
- Model rollouts per policy update: 10.
- Model rollout horizon $H_{\max}$: 1, 4, 7, 10.
- Masking rate $w$: 0.5 if $H_{\max} = 1$, else $w_h = \frac{H_{\max} - h}{2(H_{\max}+1)}$.
- Model error penalty $\alpha$: 0.001.