[Reviews · NeurIPS 2020]

Review 1

Summary and Contributions: The authors propose a novel model-based actor critic algorithm implementing a masking mechanism that avoids using the model when it is uncertain. This is motivated by the often observed reality gap when transferring a policy optimized by an inaccurate model to the real environment. Uncertainty estimation is implemented by learning an ensemble of models and doing a one vs rest comparison. Rather than measuring absolute performance, relative uncertainty across a batch of rollouts is compared and only the top n rollouts are kept, where n is a fixed hyperparameter. Comparison to state of the art alternatives is performed on 4 common MuJoCo tasks and on modified versions of these tasks, where additive white noise was introduced in the action space. They demonstrate better performance when longer rollouts are used, while longer rollouts seem to work slightly better than shorter ones.

Strengths: How to best use a model in MBRL is a key question in the current field. MBRL has become a bigger focus, rivaling model-free RL also in absolute performance. The proper usage of an inaccurate model is vital and thus, this work is very relevant to the discussion. Empirically, they compare to state of the art and add some ablation studies highlighting the proclaimed benefits of their approach. They do multiple 7 runs and report standard deviations / error bars throughout. Further, they perform some studies on their newly introduced hyperparameters, justifying their chosen values. Most convincing are the experiments on the noisy environments, where the proposed method shines as hoped by the authors. The empirical results are clearly the strongest part of the paper.

Weaknesses: The connection between the outlined theoretical argument and the proposed algorithm seems tenuous. The bound is based on a maximum error between rollouts in the real environment and in the learned dynamics, but the algorithm is based on a relative, rank-based heuristic. This is not bounding the absolute deviation in any way. The authors think this happens implicitly in 203-205, but I don't quite follow why. Even if we accept a rank-based heuristic, I’d have loved to see a more sophisticated approach to choosing w. For example, early in training one would assume larger model prediction errors, as the model has seen only few data points. Using a constant rate throughout training seems unlikely to be close to optimal, even if the value was chosen based on a hyperparamter search. In terms of novelty, mainly the algorithm/heuristic remains, as the derived bound is mainly based on prior work. Most of the derivation can be found in the appendix of MBPO.

Correctness: The claim in 100-102 is dubious. I don’t see how the bound in (2) necessarily leads to the claimed behaviour. Even if this were the case, constraining a policy update to be small in some sense (e.g. within a trust region) is one of the most researched and advocated approaches in RL. Since \policy_b is not fixed over time, I can’t follow the argument of the authors here. 311: Nothing has been shown in a real-world scenario, the claim seems a bit strong.

Clarity: I really liked the overall structure and how the reader is introduced to the problem. The nearest related work (MBPO) is appropriately covered and gives a good starting point for the discussion. Is there any explanation of non-stop mode and hard-stop mode in text? These modes' treatment, also in the ablation studies, is so short, it is possibly more confusing than helpful. Some errors: 146 Given ???? (missing) and a policy \pi 150: monotonicaly => monotonically 151: rewrited => rewritten 169: ture => true 225: earlystopping => early stopping 228: You say \sigma^2_{\theta_j} but there is no j, only i.

Relation to Prior Work: I think the similarities and differences to MBPO are mostly well laid out.

Reproducibility: No

Additional Feedback: Overall, I think this a neat heuristic with some solid evidence that it actually works. I'm torn on how to judge this submission, and may be swayed either way. If the authors could strengthen the connection between their algorithm and their theoretical analysis, it would be appreciated. Given that even high values for w work rather well (meaning only few rollouts are discarded), could the failure of model-based methods be caused by a few extreme outliers of accumulating and then possibly exploding error, resulting in a huge gradient update? This also ties in with results in Figure 3/4, where a fixed masking rate leads to very comparable results even as the environment becomes noiser (until a clearer drop off in Noisy2)? Wouldn't you have expected to use different masking rates based on your theoretical analysis, i.e. more masking in the noiser environment?


Review 2

Summary and Contributions: This paper proposes a new model-based actor critic algorithm, called Masked Model-based Actor-Critic (M2AC), that uses an estimate of the model's uncertainty combined with a masking mechanism to determine whether particular model generated experience should be used for learning or not. The paper provides general value bounds for Masked Model-based Policy Iteration, which is a central component of the algorithm, and demonstrates the effectiveness of M2AC on both deterministic and noisy continuous control problems.

Strengths: Understanding and making use of model uncertainly is an important facet of model-based RL research. This paper presents a practical approach to capture and use model uncertainty (both aleatoric and epistemic) to improve sample efficiency and performance in a practical algorithm. The paper shows that their method outperforms several model-based and model-free baselines on a number of continuous control tasks. The contribution of the idea of masked model rollouts, combined with the other components (like the One-vs-rest uncertainly estimation) is significant and sufficiently novel. This work is certainly of interest to model-based RL researchers, and likely to be of interest to the wider reinforcement learning community.

Weaknesses: In the experimental sections 5.2 and 5.3, the reported means are an average of 7 and 10 runs respectively; this feels like too few. More runs (at least 30) would significantly strengthen the results and increase my confidence in their correctness. (Why is there a difference between the number of runs in the two experiments anyways?) Additionally, I feel that some baselines are missing from section 5.3 (experiments on noisy environments). Why is M2AC only compared to MBPO? How do the model free methods perform on the noisy environments?

Correctness: See weaknesses.

Clarity: In general the paper is clearly written and easy to follow. However, I found the jump from the definition and analysis of masked model rollouts to the practical algorithm slightly confusing. Initially, it was not clear to me how masked model rollouts, whose definition has access to the true model, could be applied to a practical algorithm (the paper does state that masked model rollouts are "not a real rollout method but is useful for theoretical analysis.) Only after a careful re-read, and some additional thought, was I able to understand the connection.

Relation to Prior Work: The paper adequately discusses related work and places itself in the context of the literature.

Reproducibility: Yes

Additional Feedback: After author response period: Thank you for noting my suggestions about adding more runs and the additional baselines. My score remains the same at accept.


Review 3

Summary and Contributions: The main idea behind this paper is to expand the practical form of MBPO with localised use of simulated data for policy learning. That is, a replay buffer trick is used with the simulated data being added only for state-action pairs deemed by a "masking mechanism" to be sufficiently trustworthy. As with many other papers of this nature, authors attempt to justify their approach by a theoretical preamble, with the actual algorithm using a rather detached "bound" instead of the theoretically proven sound approximation. The "practical" form of the approach is experimentally shown to be more stable than non-localised data use of the standard MBPO and its baselines.

Strengths: The main strength is that the practical form of the proposed approach appears to be working well. The theoretical underpinning is rather overcomplicated for the end result, but it does show that a more general thinking took place to develop the idea.

Weaknesses: The end result is this: if we use simulated data for policy learning only in areas were we estimate the simulation to be faithful, we prevent the overall process to be overfitted to the phantom opportunities of the imprecise model (ensemble). It is nice to have this stated formally in a form of a theory, with distortions and partial roll-outs, and so forth, but the end algorithm is detached from that theory. Rather, it uses cross-validation (one dropout) of a model ensemble (of Gaussians) to estimate model predictive power at a locale. Simulated data is then taken with a scheduled proportion (w_h) relying on the fact that the ensemble will improve over time. This is a simple ad-hoc approach that has very little to do with the theory. It works, but it is a fairly standard approach to balance simulated and real data use. There's little novelty in it or significance.

Correctness: The empirical methodology seems to be correct, parameter sensitivity and ablation studies have been carried out for the practical algorithm. Theoretical proofs appear to be correct as well, insofar as the theory can be read, though there's little surprise in them, given the a priori very restricted and controlled simulated data injection into the estimates.

Clarity: The paper is fairly clear, with a few notable lapses that make it difficult to formally claim full clarify of theoretical portions. Theorem 2 formulation seems to be truncated by either a LaTeX or editing issue in the submitted PDF file. Some of it can be recovered from the proof, but that's no way to ensure correctness or the true intent of the authors. Quite unfortunate. What is clearly a lapse in authors attention to the theoretical part is the paragraph in lines 154-164. It is a jumble of thoughts, squashed together, and provides more of a high level intuition of a thoughts train, rather then an properly laid out interpretation.

Relation to Prior Work: Prior work is relatively well discussed.

Reproducibility: Yes

Additional Feedback: Can authors, please, elaborate on Line-10 of Algorithm-2? Specifically, the meaning of "small" when used for the uncertainty score. Line 200 of the text repeats the same argument "[portion of] samples with _small_ uncertainty". If this "small" is a parameter of the algorithm, then it should be far more explicitly discussed, controlled for and tested. As is, the suspicion is that it is the (empirical) choice of this parameter that guaranteed the use of only faithful simulation data, more than any other part of the approach. ---- post-rebuttal --- Thank you for your clarifications, they do help in reading the paper. Perhaps a more detailed and explicit description of how the practical approach follows from the theoretical motivation should be made a well visible part of the paper.


Review 4

Summary and Contributions: The paper proposes a model-based RL algorithm named Masked Model-based Actor-Critic (M2AC). They use synthetic data generated by a learned model for policy optimization (such as in MBPO), but they use a masking mechanism to only use the model-generated data when the model error is small. They derive a theoretical bound, in which the gap between the model-based and the true Q-values is bounded and can be reduced by the masking mechanism. They demonstrate state-of-the-art performance in terms of sample efficiency, and asymptotic returns comparable to state-of-the-art model-free methods but with much less real data, on 4 continuous control tasks from OpenAI Gym. In addition, their method performs consistently well with longer rollouts and in noisy environments.

Strengths: 1. They provide a novel theoretical bound of the expected return, which suggests to reduce usage of synthetic data when the model error is higher. 2. Their method achieves state-of-the-art performance in terms of sample efficiency on 4 of the continuous control benchmark environments. In addition, unlike the prior model-based method MBPO, the performance is consistent or better when using longer rollouts. 3. Their method significantly outperforms prior work on noisy versions of 2 of the environments. 4. Their method is fairly consistent across hyperparameters, and it seems that there is only a small number of hyperparameters to tune. 5. The model usage problem is a very relevant topic in model-based RL.

Weaknesses: 1. Their evaluation is performed on a small set of environments (4). The evaluation should at the very least include all the environments from the MBPO paper (namely, the inverted pendulum and ant are missing), even if shown in the appendix due to space. Additional results in other environments would make this paper more convincing. 2. Their method doesn't take into account the approximation error of the Q function on the RHS of Equation (6). Without taking that into consideration, the bound suggests that H=0 is always better (i.e. fully model-free).

Correctness: Yes.

Clarity: There are few typos throughout the paper. To name a few: Equation below line 141: h -> H Theorem 2: "Given and a policy" Line 151: rewrited -> rewritten Line 155: "an model-free return" -> "a model-free return" Line 169: ture -> true Line 173: Temperal -> Temporal Line 280: nothing -> anything

Relation to Prior Work: Yes.

Reproducibility: Yes

Additional Feedback: Section 4.2 states that one of the advantages of the masking mechanism is that "it is rank-based, we only need to find some predictor u(s,a) that is positively correlated". Although this is true for obtaining a tighter bound, is this true for the model-bias penalty? I.e. the epsilon on the second line in Equation (7).

[Author Response · NeurIPS 2020]

Thank all the reviewers for the insightful comments and the helpful suggestions!

The reviewers' common concern is about *the connections between the algorithm and the theoretical analysis*. So we
first outline the connections as follows.

1. Our return bound removes the term that involves the behavior policy, which is consistent with our algorithm. It fixes
an inconsistency issue of MBPO: their theoretical results suggest to constrain the distance between the behavior policy
and the new policy, but their algorithm does not have such a constraint, leading to bad performances in some cases.

2. The theoretical results tell us that the gap between model returns and actual returns depends on the model bias in
*model* rollouts instead of that in real rollouts (or in validation trajectories). Previous methods can be understood as
$M(s,a) \equiv 1$ for all $(s,a)$ during model rollouts though the model bias can be large (or even undefined, because the
imaginary state may not be a valid state), so they suffer from large performance gap especially when using long model
rollouts. So this result suggests to restrict model usage to reduce the gap, which leads to our actual algorithm.

3. We formulate the gap in the form of $\epsilon \cdot w$. Here $\epsilon$ is the maximum model error during *model* rollouts. So it is difficult
to quantitatively constrain $\epsilon$, if possible. However, we can always control $w$, the portion of model-generated samples to
be used. So it leads to our rank-based heuristic that selects model-generated samples with the hyper-parameter $w$.

**To Reviewer #1:**

Q1. *A more sophisticated approach to choose $w$?*

A1. We agree that it is a great idea. Actually, we have tried to choose
samples by using the samples whose predictive likelihoods are less
than the average likelihood of a hold-out validation dataset, which
had have better performance than trivial likelihood-based heuristics.
However, we do not include it because 1) we want our algorithm to
be easy-to-use, i.e., having competitive performance in most environ-
ments with the default hyper-parameters (set $w$ with a linear schedule
around 0.25), and 2) our rank-based heuristic with OvR uncertainty
estimation can have better performance.

Figure 1: A demonstration of the model rollout modes in M2AC. (a) Hard-stop mode stops model rollouts once it encounters an $(s,a)$ that $M(s,a) = 0$; (b) Non-stop mode always runs $H_{\max}$ steps and only keeps the samples that has $M(s,a) = 1$ (in green).

Q2. *Explanation of non-stop mode and hard-stop mode?*

A2. Thanks for pointing this out. As demonstrated in Figure 1, non-stop mode can provide richer samples. We will add
detailed explanation and ablation studies in text and in figure in the final version.

**To Reviewer #2:**

Q1. *7 and 10 runs are too few, at least 30 would be better. Model-free methods in the noisy environments are missing.*

A1. Thanks for the suggestion. This was due to limited computational resources (as we have to run many environments
in many different settings). We will add more runs as well as model-free baselines in the final version.

**To Reviewer #3:** Q1. *Meaning of "small" uncertainty score in Line-10 of Algorithm 2?*

A1. When the agent generates a batch of $B$ imaginary samples, it aligns an uncertainty score with the OvR estimation
for each sample. Then it ranks these samples by their uncertainty scores, and selects the first $\lfloor wB \rfloor$ samples (whose
uncertainty scores are "smaller" than others). Here, the only hyper-parameter is $w$. We show by experiments that our
default choice of $w$ works fairly well across a wide range of tasks, and the algorithm is robust with varying $w$.

Q2. *Lines 154-164 provides more of a high level intuition rather than an properly laid out interpretation.*

A2. Thanks for pointing this out. We revised this paragraph to make it clearer.

**To Reviewer #4:**

Q1. *Results should include more environments (InvertedPendulum and Ant).*

A1. Thanks for the suggestion. We will add the results in the appendix. Here
we report the average results of Ant in Table 1. As for InvertedPendulum, M2AC
performs comparably good as MBPO (Return=1000) because the task is too simple.

Table 1: Ant-v2.

| $H_{\max}$ | 1 | 4 | 7 |
|---|---|---|---|
| MBPO | 2167 | 3586 | 2394 |
| M2AC | 3907 | 4102 | 3306 |

Q2. *How to use the predictor $u(s,a)$ for the model-bias penalty?*

A2. For the model-bias penalty, since $D_{TV}(\cdot, \cdot) \leq \sqrt{D_{KL}(\cdot\|\cdot)/2}$ and our $u(s,a)$ in OvR uncertainty estimation in
Eq.(9) is a KL-divergence, we compute the sample mean of $\alpha\sqrt{u(s,a)/2}$ as the model-bias penalty. We will add
detailed explanation in the final version.

[Meta-Review · NeurIPS 2020]

The reviewers generally appreciated the paper, although Reviewer #1 raised the concern that there are some nontrivial failure cases in the algorithm, which are not explained by the theory. More generally, the reviewers were concerned about a wide theory/implementation gap. The author response clarified some of how this could be accomplished. The authors are encouraged to spend some time clarifying this issue.